# Unravelling Metabolic Heterogeneity of Chinese Baijiu Fermentation in Age-Gradient Vessels

**DOI:** 10.3390/foods12183425

**Published:** 2023-09-14

**Authors:** Zijian Miao, Yu Bai, Xinlei Wang, Chao Han, Bowen Wang, Zexia Li, Jinyuan Sun, Fuping Zheng, Yuhang Zhang, Baoguo Sun

**Affiliations:** 1Key Laboratory of Geriatric Nutrition and Health (Beijing Technology and Business University), Ministry of Education, Beijing Technology and Business University, Beijing 100048, China; 2130032042@st.btbu.edu.cn (Z.M.); by622@outlook.com (Y.B.); sunjinyuan@btbu.edu.cn (J.S.); zhengfp@btbu.edu.cn (F.Z.); sunbg@btbu.edu.cn (B.S.); 2Key Laboratory of Brewing Molecular Engineering of China Light Industry, Beijing Laboratory for Food Quality and Safety, School of Light Industry, Beijing Technology and Business University, Beijing 100048, China; 3Hebei Solid State Fermentation Making Industry Technology Research Institute, Hebei Baijiu Making Technology Innovation Center, Hebei Hengshui Laobaigan Liquor Co., Ltd., Hengshui 053000, China; wangxinlei819@163.com (X.W.); hanchao147258@outlook.com (C.H.); lizexia123@outlook.com (Z.L.); zhangyh_67@163.com (Y.Z.)

**Keywords:** fermentation vessels, microbial succession, metabolite shift, alcoholic beverages, machine learning, baijiu

## Abstract

Fermentation vessels affect the characteristics of food fermentation; however, we lack an approach to identify the biomarkers indicating fermentation. In this study, we applied metabolomics and high-throughput sequencing analysis to reveal the dynamic of metabolites and microbial communities in age-gradient fermentation vessels for baijiu production. Furthermore, we identified 64 metabolites during fermentation, and 19 metabolites significantly varied among the three vessels (*p* < 0.05). Moreover, the formation of these 19 metabolites were positively correlated with the core microbiota (including *Aspergillus*, *Saccharomyces*, *Lactobacillus*, and *Bacillus*). In addition, ethyl lactate or ethyl acetate were identified as the biomarkers for indicating the metabolism among age-gradient fermentation vessels by BP-ANN (R^2^ > 0.40). Therefore, this study combined the biological analysis and predictive model to identify the biomarkers indicating metabolism in different fermentation vessels, and it also provides a potential approach to assess the profiling of food fermentations.

## 1. Introduction

Fermented foods bring a unique flavor experience to people worldwide, and the selections of products are always determined by the safety, flavor, texture, and aroma of fermented foods [1]. The formation of foods’ quality and flavor are correlated with the metabolism of microorganisms, affected by natural or ecological factors (locality, fermentation situations or vessels, and environment) [2,3,4,5,6,7]. Therefore, it is necessary to identify the acceptable biomarkers for indicating fermentation under various natural or micro-ecology conditions in order to improve the characteristics of specific fermentation (like fermentation under different vessels).

Baijiu, as a typical spontaneously solid-state fermented alcoholic beverage, is warmly welcomed by worldwide consumers because of its unique flavor. Baijiu is usually produced by grain (sorghum, wheat, or rice), and undergoes starter production, combinations of raw materials, fermentation, distillation, and storage [8]. The flavor of various baijiu is determined by their fermentations, which are related to specific natural or micro-ecological factors, like abiotic factors (temperature, moisture, and acidity), fermentation vessels (pit mud, Digang, and stone pit), and manufacturing. Recent studies revealed the effects of environmental factors on fermentation by exploring the microbial metabolism within micro-ecology (delimited by fermented vessels) [9]. For example, the accumulations and transmissions of bio-heat are correlated with the alcoholic fermentations in the specific fermentation vessels (like Digang, a cylinder for light-aroma baijiu fermentation) [10,11,12]. Meanwhile, previous studies indicate the dynamics of microbial communities, enzymes, and metabolites under the fermentations within age-gradient vessels [13,14,15]. These studies established the correlations between metabolites and microbial communities or environmental factors to reveal the profiling of fermentations. However, identifying reasonable biomarkers for indicating the characteristics of fermentations (affected by fermentation vessels or environmental factors) remains unclear.

In this study, we selected the fermentations within age-gradient Digangs (a typical fermentation vessel for a famous baijiu production) as subjects, and combined biological analysis and a predictive model to identify the biomarker for indicating the fermentation within different vessels. First, we indicated the dynamics of metabolites and microbial communities and then established the correlations between microbial succession and metabolite shifts in age-gradient vessels (2, 15, and 30 years old). Second, we utilized a multi-algorithm predictive model to identify the biomarkers indicating fermentation within age-gradient vessels for baijiu fermentation. Our study sheds new light on the effects of micro-ecology on baijiu fermentation within a specific fermentation vessel. Meanwhile, we put forward ethyl lactate and ethyl acetate as indicators of baijiu fermentation for the first time, and also provide a potential method to identify biomarkers indicating liquor fermentation and general food fermentation.

## 2. Materials and Methods

### 2.1. Sample Collection

Jiupei (fermented grain) samples were collected from a famous distillery in Hebei province, China (37.74 N, 115.41 E). The selections of age-gradient vessels were based on the significant differences on the flavor of baijiu in practice. Three age-gradient Digangs (cylinders made of clay for baijiu fermentation, which had been used for 2, 15, and 30 years, named A, B, and C) and samples were selected at 0, 5, 10, 15, 20, and 25 days. Next, 400 g of samples (named Jiupei, fermented grain for baijiu production) were collected and stored at −80 °C for further analysis. The subsamples were collected from the upper, middle, and lower layers, and mixed as a typical sample (Figure 1). All samples were frozen immediately after collection and stored at −20 °C for analysis.

### 2.2. DNA Extraction, Amplification, and Sequence Processing

The DNA extraction and amplification of samples were described as in previous studies [16]. The barcoded PCR products were sequenced by a MiSeq benchtop sequencer for 250 bp paired-end sequencing (2 by 250 bp, Illumina, San Diego, CA, USA) at Beijing Auwigene Tech., Ltd. (Beijing, China). All sequences generated were submitted to the NCBI database under the accession numbers PRJNA956846 and PRJNA956854.

### 2.3. Metabolites Analysis

The pretreatment condition was as described in previous studies, with some modifications [17]. The metabolites were extracted and identified by headspace solid-phase microextraction (HS-SPME) coupled with gas chromatography–mass spectrometry (GC-MS) (GC 7890A-MS 5975C, Santa Clara, CA, USA). The chromatographic column was a DB-Wax (60 m × 250 μm × 0.25 μm, J&W). The carrier gas was helium with a flow rate of 1 mL/min. The temperature program was initiated at 35 °C, held for 4 min, then increased to 150 °C at a rate of 4 °C/min and held for 4 min, and finally increased to 220 °C at a rate of 3 °C/min and held for 5 min. Next, 5 g samples (Jiupei, fermented grains) were incubated at 45 °C and shaken at 450 rpm for 30 min. The extraction was operated with oscillation, at a depth of 22 mm, for 30 min, followed by an injection depth of 35 mm and desorption time of 10 s. Electron ionization (EI) mode with an ionization energy of 70 eV was used. The ion source temperature was set at 230 °C. Full-scan mode was employed in the mass range of 45 to 350 atomic mass units (AMU). The NIST 20 library was used for mass spectrometry analysis. Internal standard quantification was conducted with the addition of 100 μL of 200 mg/L internal standards, including 2-ethylbutyric acid, 4-octanol, and ethyl valerate (Aladdin, Shanghai, China), followed by calculating the content of each metabolite.

### 2.4. Establishment of Predictive Models

The data from Digang A and Digang B were classified as a training set, whereas those from Digang C were classified as a testing set. A neural network was established by MATLAB (R2022a, MathWorks, Natick, MA, USA) and SPSSPRO (v. 1.1.14), and Logistic was selected as the activation function. In order to avoid the problem of local optima, we determined the initial learning rate using an iterative method [18]. We selected the module designed by Fabian Pedregosa et al. to establish a decision tree model and support vector regression model [19], and chose the mean absolute percentage error (MAPE) and R^2^ as the evaluation parameters to evaluate the stability of three models. MAPE (a percentage value) was utilized to reflect the actual situation of prediction errors, and the smaller value indicated the higher accuracy of the model. Meanwhile, R^2^ was closer to 1 to indicate the higher accuracy of the model.

### 2.5. Olfactory Sensory Analysis

The olfactory sensory panel consisted of 12 trained panelists who were involved in olfactory experiments and quantitative description of aromas. The panelists, aged between 22 and 26, had a gender ratio of 1:1. All panelists were from the Key Laboratory of Flavor Chemistry, Beijing Technology and Business University. The sensory tests were conducted in a sensory evaluation laboratory at a temperature of 20 ± 1 °C. The baijiu samples were carefully stored, and each sample was assigned a unique identifier. Considering the optimal olfactory perception ability, the sensory evaluation was conducted from 9 am to 11 am. First, the panel members described the aroma characteristics of the baijiu samples through evaluation, and then an aroma attribute table was obtained based on the frequency and intensity of the descriptions given by the panel members. The olfactory evaluation was conducted on a six-point scale, ranging from 0 to 5. This scale helped assess the intensity of aromas, ranging from no aroma (0) to an extremely intense aroma (5).

### 2.6. Data Analysis and Availability

Figures were drafted by Origin Pro 2021 Students Edition (OriginLab Corporation, Northampton, MA, USA). The correlation network between microorganisms and metabolites was built by Cytoscape (v. 3.9.1). Statistical data were evaluated by SPSS Statistics (version 25, IBM, New York, NY, USA).

## 3. Results

### 3.1. Dynamic of Metabolites among Age-Gradient Vessels for Baijiu Fermentation

We conducted HS-SPME-GC-MS analysis to identify a total of 64 metabolites during the fermentation processes of three Digangs, namely Digang A (2 years old), Digang B (15 years old), and Digang C (30 years old). These metabolites comprised 25 esters, 7 acids, 11 alcohols, 9 phenols, and 12 others (as shown in Figure 2 and Appendix A). Specifically, Digang A exhibited the presence of 58 identified metabolites, while Digang B and Digang C had 56 and 54 identified metabolites, respectively (as shown in Appendix A). Moreover, we discovered that 48 metabolites were found in all three Digangs, accounting for approximately 75% of the total metabolites identified. This suggests a high level of stability in the metabolic profiles of these three Digangs. Notably, Digang A had two unique metabolites, namely methyl 2-methylvalerate and pelargonic acid, representing 3.1% of the total metabolites. Digang B had the unique metabolites ethyl heptanoate, propyl 2,2-dimethylvalerate, guaiacol, and furfural, accounting for 6.3% of the total metabolites. Lastly, Digang C showcased stearic acid and tetramethylpyrazine as its unique metabolites, constituting 3.1% of the total metabolites.

We used principal component analysis (PCA) to describe the dynamic of metabolites during the fermentation of the three Digangs. The changes of metabolites were mainly reflected in the first principal component axis (PC1), which accounted for 39.6% of all explained amounts during the fermentation of the Digangs. PC1 was mainly related to fermentation time, indicating that fermentation time was the most important factor affecting the metabolism of Digang fermentation. PC2 mainly indicated the effect caused by container difference, which accounted for 21.8% of the metabolic difference during the fermentation (Figure 2a). In addition, the comparison showed that the kinds of metabolites had high consistency; however, the compositions of metabolites showed significant differences in the three Digangs, especially the 19 identified differential metabolites (12 esters, 5 alcohols, 1 phenol, and 1 ketone) (Figure 2).

#### 3.1.1. Dynamic of Esters

A total of 25 esters were detected in three different Digangs. Among them, 23 esters were determined in Digang A, 22 esters in Digang B, and 19 esters in Digang C. According to the national standard of Chinese baijiu, the ratio of ethyl lactate and ethyl acetate in Laobaigan baijiu (the specific baijiu made in the vessel named Digang for this study) should be above 0.8 for strengthening the floral and fruit aromas of liquor, so the compositions of ethyl lactate and ethyl acetate are important indexes to measure the excellence of Laobaigan baijiu [20,21]. We further found that the metabolic activity of ethyl lactate, ethyl acetate, 4- ethylguaiacol, and 3-hydroxy-2-butanone gradually increased, with the aging of Digang (from 2 to 30 years old) for enhancing the aroma characteristics of Laobaigan baijiu.

During the fermentation, the content of ethyl acetate gradually increased during the fermentation of Digang, from 22.35 mg/kg to 297.07 mg/kg in Digang A, 19.68 mg/kg to 234.95 mg/kg in Digang B, and 22.35 mg/kg to 218.92 mg/kg in Digang C. At the end of fermentation, the content of ethyl acetate fluctuated greatly with the ages of the Digangs (*p* < 0.05). The metabolic activity of ethyl acetate gradually decreased from Digangs A to C with the aging of vessels (*p* < 0.05). In contrast, the metabolic activity of ethyl lactate gradually increased from Digang A to B and C, and the contents of ethyl lactate were 246.13 mg/kg, 255.94 mg/kg, and 254.88 mg/kg at the end of fermentation in Digangs A, B, and C (Appendix A). Therefore, the ratios of ethyl lactate and ethyl acetate increased to 0.83, 1.09, and 1.16 with the aging of vessels (from Digang A to B and C) at the end of fermentation.

#### 3.1.2. Dynamic of Volatile Organic Acids

A total of seven volatile organic acids were identified in the fermentation of the three Digangs (Appendix A). Among them, six acids were determined in Digang A, five acids in Digang B, and six acids in Digang C (Appendix A). In general, the organic acids were rapidly accumulated in the early stage of fermentation (0–5 days), especially the metabolism of acetic acid. The metabolic activity of acetic acid in Digang C (27.74 mg/kg) was significantly lower than that in Digang B (41.29 mg/kg) and Digang A (36.32 mg/kg) (*p* < 0.01). In contrast, the metabolism of caproic acid was enhanced from 0 to 0.87 mg/kg, 0.97, and 0.83 mg/kg, whereas that of octanoic acid was enhanced from 0 to 1.26 mg/kg, 1.00 mg/kg, and 0.47 mg/kg with the aging of vessels (from Digang A to B and C).

#### 3.1.3. Dynamic of Alcohols

A total of 11 alcohols were identified in the fermentation of the three Digangs. Among them, 11 alcohols were determined in Digang A, 10 alcohols in Digang B, and 10 alcohols in Digang C (Appendix A). The production of ethanol mainly occurred in the early stage of fermentation (0–10 days), and then decreased in the late stage of fermentation (10–25 days) that might be transformed to esters. In the end of fermentation, the content of ethanol increased from 25.09 mg/kg, 26.73 mg/kg, and 25.09 mg/kg to 1597.59 mg/kg, 1529.28 mg/kg, and 1835.31 mg/kg with the aging of the vessels (from Digang A to B and C).

In addition, we mainly identified phenylethanol, isoamyl alcohol, and 2,3-butanediol in the fermentation of Digang, which might add a bitter and spicy taste to liquor body and cause discomfort after drinking [22]. The content of higher alcohols in the fermentation of Digang C was generally lower than that in Digang B and Digang A, especially phenyl ethanol and 2,3-butanediol (Appendix A). At the end of fermentation, the contents of phenylethanol and 2,3-butanediol were 69.20 mg/kg and 9.78 mg/kg in Digang C, which were lower than those in Digang B (88.99 mg/kg and 14.98 mg/kg) and Digang A (91.63 mg/kg and 16.35 mg/kg).

#### 3.1.4. Dynamic of Phenols

Nine phenols were identified within these fermentations of the three Digangs (Figure 2b and Appendix A). Eight phenols were determined in Digang A, nine phenols in Digang B, and eight phenols in Digang C. In particular, the content of 4- ethylguaiacol in Digang C (3.26 mg/kg) was 6.37 times and 4.65 times that of Digang A (0.52 mg/kg) and Digang B (0.70 mg/kg) in the late stage of fermentation, respectively. Therefore, this indicated that the metabolism of 4- ethylguaiacol increased with the ages of Digangs (from Digang A to B, C), which contributes an effective natural antioxidant to Chinese baijiu (Zhao et al., 2017).

### 3.2. Aroma Profile Analysis of Metabolites in Three Digangs

Grain, cooked apple, fruity, floral, aged, grass, acidic and alcoholic were selected as odor descriptors, and the sensory scores were significantly different among different liquor samples (*p* < 0.05, Appendix A). In addition, with the increase of fermentation vessel age, cooked apple, fruity, floral and aged descriptors showed higher intensity, which was consistent with the quantitative results of metabolites.

### 3.3. Microbial Successions in the Fermentation of Three Digangs

We utilized a high-throughput sequencing technique to reveal microbial diversity and succession during the fermentation of the three Digangs. A total of 350,676 clean reads were obtained from bacterial sequencing results and 330,000 clean reads were obtained from fungal sequencing results. The good coverage rates of all samples were above 0.99, ensuring the reliability of the data (Appendix A).

We found higher bacterial or fungal diversity in Digangs A and C (Appendix A). In the front and middle stages of fermentation (0–15 days), the bacterial chao1 index increased to 65.14, and then decreased to 50.43 in Digang A. Meanwhile, the bacterial chao1 index rapidly decreased from 74.00 to 8.25 in Digang B, and the bacterial chao1 index rose to 32.67 in Digang C, which was significantly lower than that in Digang A. On the contrary, the fungal chao1 index decreased from 28.2 and 28 to 17.5 and 7 in Digang A and Digang B, whereas the fungal chao1 index increased from 35.2 to 36 in Digang C.

For bacterial communities, 180 genera were identified in three Digangs, including 130 in Digang A, 72 in Digang B, and 49 in Digang C. For fungal communities, we identified 36 fungal genera in the three Digangs, including 29 genera in Digang A, 25 in Digang B, and 27 in Digang C (Appendix A).

We applied principal component analysis to reveal the dynamic of microbial communities among these three vessels (Figure 3b,d). PC1 and PC2 accounted for 91% of the information for bacterial communities and 98.6% for the fungal communities during fermentation (Figure 3). The dynamic of microbial communities showed the differential trails among these three Digangs, especially in the early stage of fermentation (0–10 days) (Figure 3b,d).

In the process of fermentation, the microorganisms with relative abundance over 1% were mainly Saccharomyces, Kazachstania, Lichtheimia, Wickerhamomyces, Aspergillus and Lactobacillus, Pseudomonas, Staphylococcus, Bacillus, Weissella, Leuconostoc, Kroppenstedtia, and Pediococcus (Figure 3). There were several microorganisms which varied between the three Digangs, especially Lactobacillus, Saccharomyces, Pseudomonas, Bacillus, Lichtheimia, and Kazachstania (*p* < 0.05). On the 15th day of fermentation, the relative abundance of Lactobacillus decreased to 1.01% in Digang A, while it increased to 99.68% and 68.97% in Digang B and Digang C, respectively. However, the relative abundance of Saccharomyces reached 95.78%, 97.35%, and 69.08% on the 15th day of fermentation, respectively, with the aging of the vessels (from Digang A to B and C). In addition, these differences were also found in the growth of Pseudomonas, Bacillus, Lichtheimia, and Kazachstania (in the stage of 0–15 days of fermentation) among the three Digangs (Figure 3).

### 3.4. Correlations between Metabolites and Microbial Communities during the Fermentation of Three Digangs

We established the correlation network and microbial co-occurrence network diagram with the Spearman coefficients between microorganisms or microorganisms and metabolites (Figure 4). We found 29 significant correlations between microbes (R^2^ > 0.6, *p* < 0.05), including 9 positive correlations and 20 negative correlations (Figure 4a). Furthermore, we found the more positive correlations among the dominant genera, especially *Saccharomyces*, *Kazachstania*, *Lichtheimia*, *Wickerhamomyces*, *Aspergillus* and *Lactobacillus*, *Pseudomonas*, *Staphylococcus*, *Bacillus*, *Weissella*, *Leuconostoc*, *Kroppenstedtia*, and *Pediococcus* (R^2^ > 0.6). For example, *Bacillus* was the active genus that had the most positive correlations with other microorganisms, including *Pichia*, *Staphylococcus*, *Weissella*, *Methyloversatilis*, *Leuconostoc*, *Kroppenstedtia*, *Pediococcus*, and *Lichtheimia* (Figure 4a). Therefore, theses interactions might affect the succession and assembly of microbial communities for driving the metabolite shifts in baijiu fermentation [23].

In addition, we also found 91 significant correlations between metabolites and microorganisms, including 27 positive and 64 negative correlations (Figure 4b,c). For example, the bacterial genii *Lactobacillus*, *Kroppenstedtia*, *Weissella*, and *Pediococcus* were positively correlated with the formation of 2-Methyl-1-propanol, 3-(Methylsulfanyl)-1-propanol, Ethyl lactate, Ethyl caprate, Diethyl succinate, Ethyl 2-hydroxy-4-methylvalerate, and 3-Methyl-2(5H)-furanone (R^2^ > 0.5) (Figure 4b). Meanwhile, the fungal genera (*Saccharomyces*, *Pichia*, *Wickerhamomyces*) were positively correlated with the formation of Phenethyl alcohol, 3-(Methylsulfanyl)-1-propanol, Ethyl acetate, Ethyl caprylate, Ethyl palmitate, Ethyl laurate, Ethyl lactate, Ethyl caprate, Diethyl succinate, and 3-Methyl-2(5H)-furanone (R^2^ > 0.5) (Figure 4c). Especially, the dominated *Saccharomyces* and *Lactobacillus* were positively correlated with the metabolism of ethyl lactate, ethyl acetate, and ethanol (Figure 4).

## 4. Multi-Algorithm Prediction Aided Verification of Metabolic Correlation

### 4.1. Establishment of Neural Network Model

We predicted the metabolism of microbial communities in age-gradient fermentation vessels with machine learning (Figure 5). Artificial neural networks (ANNs) are common mathematical models for handling and predicting the metabolism of complex microbial communities, especially the algorithm of back propagation (BP) [24,25]. In this study, we built a BP-ANN model based on microbial communities and metabolites (Figure 1 and Figure 3). We sent the contents of the dominant fungi and bacteria (located in the top 10) into the algorithm, and set the metabolites as output values (Figure 5c). After testing 19 key metabolites (differently expressed among the fermentation of the three Digangs) (Appendix A), the prediction curves of ethyl lactate and ethyl acetate had the best fitting degree (R^2^ > 0.40). Therefore, ethyl lactate and ethyl acetate were chosen as anchor points to evaluate the stability and accuracy of the predictive model.

### 4.2. Evaluation of Neural Network Model

The MAPE values of the models were 4.685 (ethyl lactate) and 3.137 (ethyl acetate) on the training set. R^2^ values were 0.981 (ethyl lactate) and 0.960 (ethyl acetate) on the training set. The test data showed the good predictive accuracy of the BP-ANN model, indicating that the prediction values were close to the true value and stable. Therefore, the trained BP-ANN model was used to predict the metabolism, and the MAPE values were 26.517 (ethyl lactate) and 15.925 (ethyl acetate) on the testing set. R^2^ values were 0.645 (ethyl lactate) and 0.481 (ethyl acetate) on the training set.

### 4.3. Construction and Evaluation of Other Predicting Models

#### 4.3.1. Construction and Evaluation of Decision Tree Model

Decision tree (DT) is a tree structure model that consists of nodes and directed edges for optimizing fermentation process parameters and predict biological sources [26,27] (Figure 5a). The decision pathway of the model starts from the root node, and performs decision analysis in the following sub-nodes. Tests are accorded to a certain feature of input data, and then split the results to the next level child nodes. The input samples are repeatedly calculated, and finally the results are obtained at the leaf nodes [28]. In this study, the data of microorganisms and metabolites were imported into the DT model for training and testing, and the MAPE values of the DT model were 19.121 (ethyl lactate) and 25.051 (ethyl acetate) on the testing set. R^2^ values were 0.508 (ethyl lactate) and 0.289 (ethyl acetate) on the test set.

#### 4.3.2. Construction and Evaluation of Support Vector Regression Model

Support vector regression (SVR) is a regression model that is widely used to estimate the nutrient content in foods, built by a support vector machine [29]. This model serves the purpose of mapping the input data into a high-dimensional feature space, then finds a hyperplane in this space so that it can contain as much input data as possible, and the nonlinear mapping ability is mainly provided by its kernel function [30] (Figure 5b).

In this study, we selected Linear as the kernel function. The data of microorganisms and metabolites were imported into the model for training and testing, and the MAPE values of the model were 33.395 (ethyl lactate) and 19.477 (ethyl acetate) on the testing set. R^2^ values were 0.604 (ethyl lactate) and 0.453 (ethyl acetate) on the testing set.

### 4.4. Comparison of Different Predicting Models

We compared the predictability of the three models, and BP-ANN showed better applicability compared with the other predictive models (SVR, DT) (Appendix A). This BP-ANN model could establish a reliable relationship between input characteristic parameters and output prediction results, and might be more suitable for metabolic prediction of complex microbial communities and be used as a powerful auxiliary verification to traditional correlation analysis methods.

## 5. Discussion

In this study, we revealed the dynamics of metabolites and microbial communities among three age-gradient vessels (2, 15, and 30 years old) for baijiu fermentation. Furthermore, we established a predictive model to identify the biomarker for indicating the metabolism within age-gradient vessels.

### 5.1. Microbial Succession Drove Metabolite Shifts in Age-Gradient Vessels

We identified 64 metabolites during the fermentations of the three Digangs, and 48 metabolites were metabolized by all three Digangs, accounting for 75% of the total, indicating that the production of the three Digangs was stable (Appendix A). Although the kinds of metabolites had high consistency, the contents of metabolites showed significant differences among these three Digangs, especially the 19 identified differential metabolites (including 12 esters, 5 alcohols, 1 phenol, and 1 ketone), especially acetic acid, ethyl lactate, ethyl acetate, and 4-ethylguaiacol (Figure 2).

Ethanol, acetic acid, and lactic acid are the precursors of ethyl acetate and ethyl lactate, and are always metabolized by the species of *Lactobacillus* and *Saccharomyces* in baijiu fermentation [31,32,33]. In this study, we found that the metabolism of ethyl lactate gradually increased, while ethyl acetate decreased from Digangs A to B and C with the aging of vessels from 2 to 15 and 30 years old (Appendix A, Figure 2). This tendency significantly corresponded with the dynamic of *Lactobacillus* and *Saccharomyces* among the three Digangs, especially in the early stage of fermentation (0–15 days) (Figure 3 and Figure 4). In addition, the co-culture of *Saccharomyces* and *Lactobacillus* could enhance the formation of esters in baijiu fermentation [34,35]. We also observed a consistent correlation between the microbiota and metabolites through our calculations and simulations, which aligned with previously studied microbial metabolic pathways, particularly in the metabolism of *Saccharomyces*, *Lactobacillus*, and ethyl lactate and acetic acid [36,37,38]. Furthermore, the formation of 4-ethylguaiacol was enhanced with the aging of vessels, whose formation was positively correlated with *Bacillus*, consistent with the growth of *Bacillus* from Digangs A to B and C (2 to 15 and 30 years old) (Appendix A, Figure 4) [39]. Therefore, this indicated that microbial succession drove the metabolite shifts among age-gradient vessels, and the growing of microorganisms enhanced the formation of metabolites with the aging of vessels (from 2 to 15 and 30 years old).

The succession was mainly driven by specific microorganisms, including *Saccharomyces*, *Kazachstania*, *Lichtheimia*, *Lactobacillus*, *Bacillus*, *Weissella*, and *Pediococcus* (Figure 2 and Figure 4). In particular, these changes mainly occurred in relation to the growth of fungi (*Saccharomyces*, *Kazachstania*, *Lichtheimia*), whose growth is necessary for the requirement of certain oxygen [40]. The effects of the vessel’s material on the porosity were also considered in the discussion on variations between fermentation under age-gradient vessels in our study. The vessel undergoes the stress of acidity and moisture from the fermentation of baijiu with the extension of utilization. These extreme environments may affect the porosity of vessels and induce the variations in oxygen and bio-heat transmission (Appendix A). Subsequently, the variations in oxygen and bio-heat transmission might drive the succession of the microbial community and affect the metabolism, such as by activating the growth of fungi in the early stage of fermentation. The enhanced growth of molds (*Aspergillus* and *Lichtheimia*) may promote the activity of glucoamylase and hydrolyze the polysaccharides to the fermentable sugars [41]. Furthermore, the accumulation of fermentable sugars may promote the growth of *Saccharomyces*, flavor-producing bacteria (*Lactobacillus*, *Bacillus*, etc.), or their interactions, for improving the formation of metabolites (like ethanol, lactic acid, and acetic acid) in baijiu fermentations [42]. The primary metabolites could be subsequently transformed into ethyl lactate, ethyl acetate, ethyl laurate, and so on, which contribute to the flavor of Laobaigan baijiu [42,43].

### 5.2. Ethyl Lactate and Ethyl Acetate Are Indicators of the Fermentation Process

We indicated that microbial succession drove the metabolite shift in age-gradient fermentation vessels, strengthening the effects of vessels on the qualities of food fermentations [10]. However, we still lack an approach to identify the biomarkers indicating the metabolism of food fermentations based on microbial communities and metabolites. In this study, we identified 19 metabolites that significantly varied among the three vessels (*p* < 0.05). In addition, the dynamic of these metabolites was significantly related to the succession of microbial communities among three vessels, ethyl lactate and ethyl acetate in particular (Figure 4). Furthermore, we attempted to establish three models to predict the formation of metabolites based on the compositions of microbial communities (Figure 5). We tried to predict the metabolism of 19 metabolites (differently expressed among the three vessels) based on the compositions of the dominant microbiota (located in the top 10), to identify the suitable biomarker for indicating the profiling of fermentations (Figure 5 and Appendix A). Moreover, we found that BP-ANN was more stable and valid in predicting metabolites of complex microbial communities through the assessment and comparison of multiple algorithms (Figure 5). Furthermore, the prediction curves of ethyl lactate and ethyl acetate had the best fitting degree (R^2^ > 0.40) above the other 17 metabolites (Appendix A). In addition, ethyl lactate and ethyl acetate are the key flavor compounds that bring fruit fragrance to baijiu, always serving as a key factor for evaluating the quality and flavor of baijiu [44]. Therefore, ethyl lactate and ethyl acetate might be suitable as the biomarkers for indicating the characteristics of baijiu fermentations, based on biochemistry, prediction, and flavor contribution analysis (Figure 4). The formations of ethyl lactate and ethyl acetate were positively correlated with the core microorganisms (like *Saccharomyces*, *Lactobacillus*, *Bacillus*, etc.), and the variations of metabolites might illustrate the effects of vessels on fermentations. Furthermore, the efficient prediction of ethyl lactate and ethyl acetate indicated that the ANN-BP model might be universal for the metabolic prediction of complex microbial communities. It also provided a potential approach to identify the biomarkers indicating the metabolism of baijiu fermentation in particular and other food fermentations in general.

## 6. Conclusions

In this study, we indicated the dynamics of metabolites and microbial communities among three age-gradient vessels (2, 15, and 30 years old) for baijiu fermentation. Furthermore, we established a predictive model to identify the biomarker for indicating the metabolism under different micro-ecology. Meanwhile, we strengthened BP-ANN as a suitable metabolic predictor in complex microbial communities. Therefore, we indicated the effects of micro-ecology (fermentation vessels) on baijiu fermentation. Meanwhile, we revealed ethyl lactate and ethyl acetate as indicators of baijiu fermentation, considering that the ratio of ethyl lactate to ethyl acetate at the end of fermentation should be not less than 1.0 as an indicator of excellent fermentation. This also provides a potential approach to identify the biomarkers for indicating baijiu fermentation in particular and food fermentations in general.

## Figures and Tables

**Figure 1 foods-12-03425-f001:**
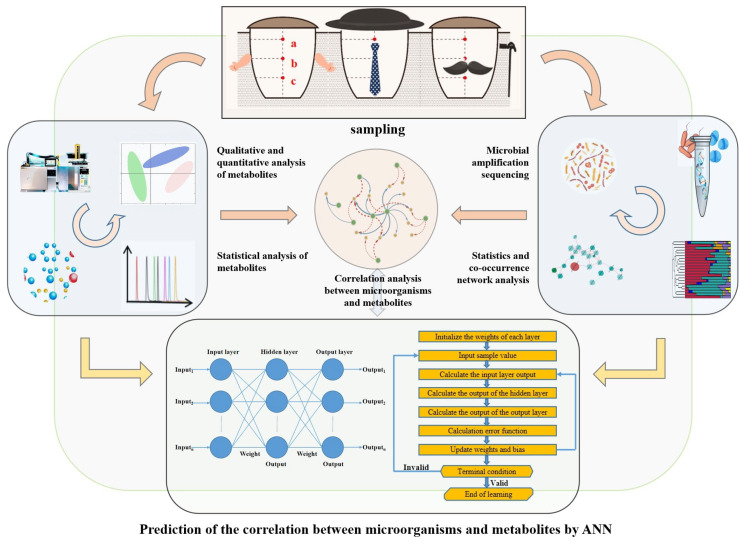
Spectra of experimental designs and methods used in this study.

**Figure 2 foods-12-03425-f002:**
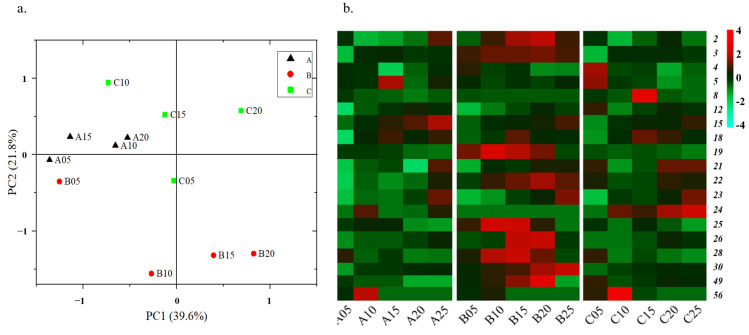
Dynamic of metabolites among the fermentation of three vessels. (**a**) PCA analysis based on metabolites. (**b**) Heat map of different metabolites metabolized among three Digangs A, B, and C. 2: Phenethyl alcohol, 3: 3-Methyl-1-butanol, 4: 2-Methyl-1-propanol, 5: 2,3-Butanediol, 8: 3-(Methylsulfanyl)-1-propanol, 12: Ethyl oleate, 15: Ethyl acetate, 18: Ethyl caprylate, 19: Ethyl valerate, 21: Ethyl palmitate, 22: Ethyl laurate, 23: Ethyl lactate, 24: Ethyl nonanoate, 25: Ethyl hexanoate, 26: Ethyl caprate, 28: Ethyl butanoate, 30: Diethyl succinate, 49: 4-Ethylguaiacol, 56: 3-Hydroxy-2-oxobutane.

**Figure 3 foods-12-03425-f003:**
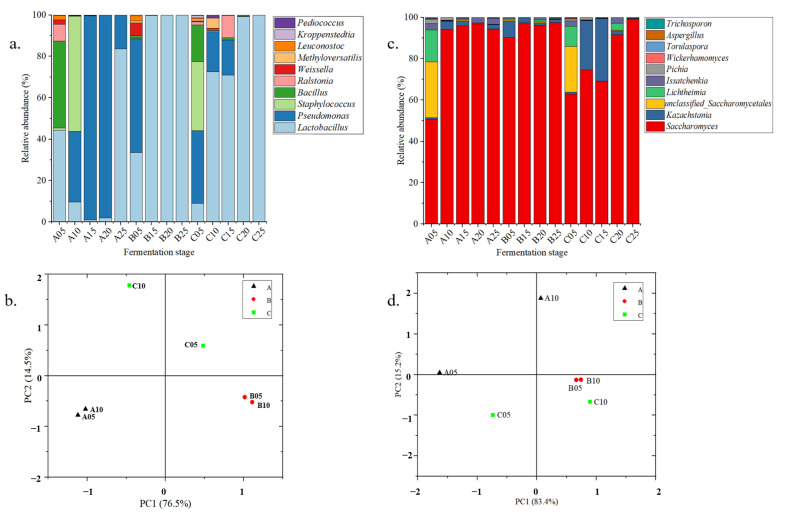
Microbial structures and succession among the fermentations of three vessels. (**a**) Compositions of the dominated genera of bacteria. (**b**) PCA of the dominated bacteria in the early fermentation stage (0–10 days). (**c**) Compositions of the dominated genera of fungi. (**d**) PCA of the dominated fungi in the early fermentation stage (0–10 days).

**Figure 4 foods-12-03425-f004:**
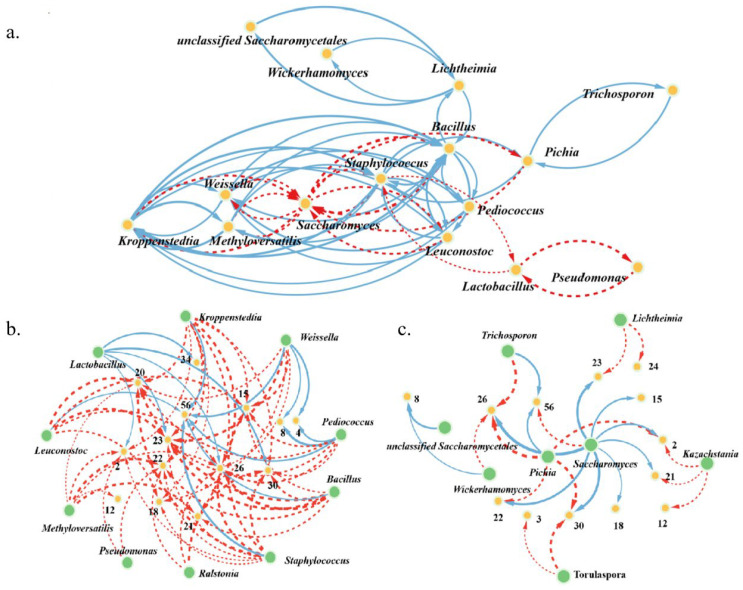
Co-occurrence network between microorganisms or microorganisms and metabolites. (**a**) Microbial co-occurrence network diagram. (**b**) Correlation diagram between bacteria and metabolites. (**c**) Correlation diagram between fungi and metabolites. The blue solid line represents positive correlation, the red dotted line represents negative correlation, and the thickness of the line represents the degree of correlation. The thicker the lines, the stronger the correlations.

**Figure 5 foods-12-03425-f005:**
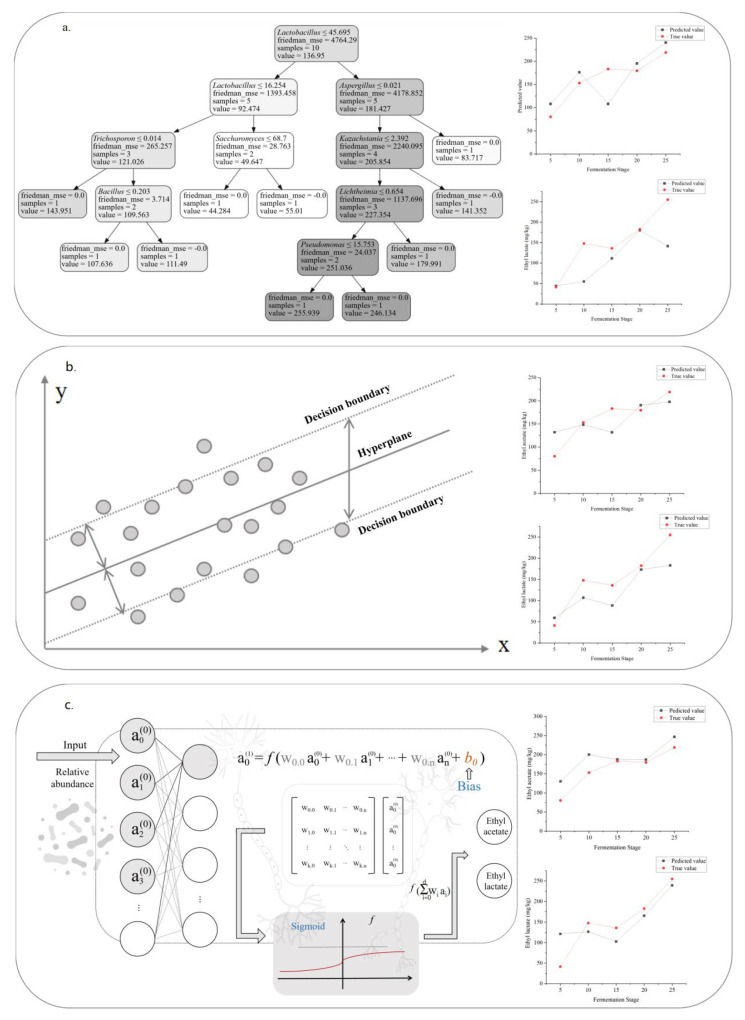
Metabolic prediction of ethyl lactate and ethyl acetate by different models. (**a**) Decision tree (DT). (**b**) Support vector regression (SVR). (**c**) Back propagation–artificial neural network (BP-ANN).

## Data Availability

The data used to support the findings of this study can be made available by the corresponding author upon request.

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
