# Peer review of "Unravelling Metabolic Heterogeneity of Chinese Baijiu Fermentation in Age-Gradient Vessels"

_foods, 2023, doi:10.3390/foods12183425_

Round 1

Reviewer 1 Report

The manuscript describes metabolomic and microbial community analysis of baijiu fermentation and creates new models for future predictions.

The introduction is quite repetitive, however, there are important characteristics of the processes that are not explained making difficult to understand, at least for this reviewer, the experimental design:

- It is not clear if the fermentation process is spontaneous or it is inoculated with a  previously fermented product

- The fermentation process is extended until 30 years or more? For example, the expression “the front stage of fermentation” what does it mean? Is a reference about time or position in the vessel?

- Why do you take the samples in this time points independently the time of the vessels?

- Why do you use this three times? Figure 1 could be a nice opportunity to clarify.

- The knowledge obtained by this fermentation process is really useful to understand other fermentative processes as you claim in introduction and conclusion? Why?

Also, the units used for metabolites concentration are expressed in mg/kg so you refer a solid but you don’t explain it is a solid fermentation, you do the SPME from a solid, I could not find data about the size of the sample and internal standard. There is a bibliographic reference instead, but at least this basic information should be given in the manuscript.

Author Response

尊敬的审稿人 1
我们非常感谢您的意见和建议。这些建议对我们改进学习非常有帮助。我们重新提交了一份题为“揭示中国白酒发酵在年龄梯度容器中的代谢异质性”的新手稿(原始编号Foods-2584307)。我们对您意见的答复已附后,修订的部分在新稿件中用不同的颜色(红色)标记,以突出显示修订。
如果您有任何问题,请随时与我们联系。我们真诚地希望这份手稿可以考虑在《食品》杂志上发表。再次感谢您的所有帮助。
致以最诚挚的问候,

博文 通讯作者

Reviewer 2 Report

GENERAL COMMENTS

This manuscript reports using metabolomics and high-throughput sequencing analysis to reveal the dynamic of metabolites and microbial communities in age-gradient fermentation vessels for baijiu production. The authors identified 64 metabolites during fermentations, and 19 metabolites significantly varied among the vessels used. They claim that these 19 metabolites' formation positively correlated with the core microbiota (including Aspergillus, Saccharomyces, Lactobacillus, and Bacillus). Ethyl lactate or ethyl acetate was identified as the possible biomarker for indicating the metabolism among age-gradient fermentation vessels. The authors concluded that combining the biological analysis and predictive model can allow the identification of biomarkers for predicting the metabolism in different fermentation vessels.

The work is interesting, the methodology is appropriate, and the conclusion is well-supported. English is generally understandable, but a careful review of the wording is necessary. There are many grammatical and punctuation errors throughout the manuscript. Specific comments 2-11 give some examples. It is recommended that a native English speaker review the manuscript. Some technical comments are the following.

  1. The introduction needs to include documented information on the effect of fermentation vessel age. Where does the hypothesis that it impacts fermentation and product quality come from?
  2. It is understood that the work was pervasive since it included many determinations to identify strains and metabolites and try to correlate them. However, sampling just one fermentation vessel of each age is enough to claim "the contents of metabolites showed significant differences among these three Digangs"? (Lines 324 – 325). To speak of significant differences, at least two vessels of each age should have been sampled. Please argue about it in the discussion section.
  3. The authors identify ethyl lactate and ethyl acetate as indicators of fermentation. However, the manuscript does not state how or to what extent those metabolites play that role. What concentration of each indicates that fermentation is occurring correctly? or what relationship between them is an indicator of a good fermentation? How does this depend on the age of the fermentation vessel? Please include it in the discussion section.
  4. Finally, the conclusion of the manuscript should include a statement regarding how the identified biomarkers indicate proper fermentation progress. It should be quantitative, establishing biomarker concentrations or relationships between them that indicate adequate fermentation.

SPECIFIC COMMENTS

  1. Line 12. Fermentation vessel is not the only factor determining the characteristics of food fermentations. At least, it can't say that forcefully. Please remove this affirmation.
  2. Line 28. It is missing an article "a" before the word "unique." The phrase "all over the world" may be wordy. Please consider using only the word "worldwide."
  3. Line 29. It is missing a comma before the words "and aroma of."
  4. Line 32. "Environmental" seems to fix this context better than "environment." Using "etc." in formal writing is generally frowned upon. Please consider rewriting the sentence.
  5. Line 34. It seems that there is an unnecessary comma (micro-ecology,"). Consider removing the comma.
  6. Line 37. The use of the preposition "as" may be incorrect here. Please consider using "because of" rather than "as".
  7. Line 39. It is missing a comma before the words "and acidity." It is missing a comma before the words "and manufacturing."
  8. Line 41. The use of the preposition "through" may be incorrect here. Please consider using "by" rather than "through."
  9. Line 46. The phrase "the identification of" may be wordy. Please consider using only the word "identifying."
  10. Line 47. The word "characteristic" may not agree in number with other words in this phase." Please use "characteristics" rather than "characteristics."
  11. Line 48. "Still remains" may be redundant. Please remove the word "still".
  12. Line 49 – 59. Please provide the volume of the fermentation vessels and the volume of each sample obtained.
  13. Line 72. The reference "(Avershina, Frisli & Rudi, 2013)" is mentioned differently than others in the text.
  14. Line 78. The reference [12] is on mathematical modeling of similar fermentation systems used here, but it does not describe any extraction of metabolites. Please correct it.
  15. Line 78 – 83. Please provide more details on metabolite identification: injector temperature, carrier gas and flow used, detector temperature, and ionization voltage. Were compounds identified by comparing the mass spectra with those in the NIST database? Did the authors use the Kovats index or standard compound injections to reinforce the identification of compounds?
  16. Line 144 – 133. The methodology described (partially) in the subsection "2.3. Metabolite analysis" only allows us to determine the presence of the metabolites. Please explain the procedure to quantify the metabolites in the Materials and Methods section.
  17. Line 328 – 341. The correlation made in these lines between metabolites and strains must be complemented with arguments, including the metabolic capacities of the identified strains. It may allow an explanation of the associations made between metabolites and microbial strains.
  18. Line 346 – 348. Do the authors consider that vessel material progressively increases its porosity or something like that? Please explain the reason for their affirmation.
  19. Line 360. The reference "(Gil I Cortiella et al., 2021)" is mentioned differently than others in the text.

English is generally understandable, but a careful review of the wording is necessary. There are many grammatical and punctuation errors throughout the manuscript.

Author Response

Dear Reviewer 2
We appreciated very much for comments and suggestions from you. These suggestions were very helpful for us to improve our study. We have resubmitted a new manuscript entitle “Unravelling metabolic heterogeneity of Chinese baijiu fermentation in age-gradient vessels” (original number of Foods-2584307). Our reply to your comments has been attached, and the revised parts were marked in a different color (red) in the new manuscript to highlight revisions.
Please do not hesitate to contact us if you have any questions. We sincerely hope this manuscript could be considered for publication on Foods. Thanks again for all your help.
With best regards, 

Bowen Wang 
Corresponding authors

Reviewer 3 Report

Dear authors, I have analyzed the manuscript and you may find my observations attached to the text highlighted in yellow.

I recommend major revision of the paper!

Author Response

Dear Reviewer 3
We appreciated very much for comments and suggestions from you. These suggestions were very helpful for us to improve our study. We have resubmitted a new manuscript entitle “Unravelling metabolic heterogeneity of Chinese baijiu fermentation in age-gradient vessels” (original number of Foods-2584307). Our reply to your comments has been attached, and the revised parts were marked in a different color (red) in the new manuscript to highlight revisions.
Please do not hesitate to contact us if you have any questions. We sincerely hope this manuscript could be considered for publication on Foods. Thanks again for all your help.
With best regards, 

Bowen Wang 
Corresponding authors

Round 2

Reviewer 2 Report

The authors have answered all my questions and suggestions. I have no further comments.

Reviewer 3 Report

Accept in present form.